# Novel Megaplasmid Driving NDM-1-Mediated Carbapenem Resistance in *Klebsiella pneumoniae* ST1588 in South America

**DOI:** 10.3390/antibiotics11091207

**Published:** 2022-09-07

**Authors:** Mario Quezada-Aguiluz, Andrés Opazo-Capurro, Nilton Lincopan, Fernanda Esposito, Bruna Fuga, Sergio Mella-Montecino, Gisela Riedel, Celia A. Lima, Helia Bello-Toledo, Marcela Cifuentes, Francisco Silva-Ojeda, Boris Barrera, Juan C. Hormazábal, Gerardo González-Rocha

**Affiliations:** 1Laboratorio de Investigación en Agentes Antibacterianos (LIAA-UdeC), Facultad de Ciencias Biológicas, Universidad de Concepción, Concepción 4030000, Chile; 2Departamento de Medicina Interna, Facultad de Medicina, Universidad de Concepción, Concepción 4030000, Chile; 3Millennium Nucleus for Collaborative Research on Bacterial Resistance (MICROB-R), Santiago 8320000, Chile; 4Centro Regional de Telemedicina y Telesalud del Biobío (CRT Biobío), Concepción 4030000, Chile; 5Department of Clinical Analysis, School of Pharmacy, University of São Paulo, São Paulo 05508-000, Brazil; 6Department of Microbiology, Institute of Biomedical Sciences, University of São Paulo, São Paulo 05508-000, Brazil; 7Unidad de Infectología, Hospital Regional “Dr. Guillermo Grant Benavente”, Concepción 4030000, Chile; 8Servicio de Laboratorio Clínico, Hospital Clínico Universidad de Chile, Santiago 8320000, Chile; 9Subdepartamento de Enfermedades Infecciosas, Instituto de Salud Pública de Chile (ISP), Santiago 8320000, Chile

**Keywords:** NDM-1, carbapenem-resistant Enterobacterales, *Klebsiella pneumoniae*, plasmid transfer, carbapenemases

## Abstract

Carbapenem-resistant Enterobacterales (CRE) is a critical public health problem in South America, where the prevalence of NDM metallo-betalactamases has increased substantially in recent years. In this study, we used whole genome sequencing to characterize a multidrug-resistant (MDR) *Klebsiella pneumoniae* (UCO-361 strain) clinical isolate from a teaching hospital in Chile. Using long-read (Nanopore) and short-read (Illumina) sequence data, we identified a novel un-typeable megaplasmid (314,976 kb, pNDM-1_UCO-361) carrying the *bla*_NDM-1_ carbapenem resistance gene within a Tn*3000* transposon. Strikingly, conjugal transfer of pNDM-1_UCO-361 plasmid only occurs at low temperatures with a high frequency of 4.3 × 10^−6^ transconjugants/receptors at 27 °C. UCO-361 belonged to the ST1588 clone, previously identified in Latin America, and harbored aminoglycoside, extended-spectrum β-lactamases (ESBLs), carbapenem, and quinolone-resistance determinants. These findings suggest that *bla*_NDM-1_-bearing megaplasmids can be adapted to carriage by some *K. pneumoniae* lineages, whereas its conjugation at low temperatures could contribute to rapid dissemination at the human–environmental interface.

## 1. Introduction

Carbapenem-resistant Enterobacterales (CRE) is a global concern for public health due to the limited therapeutic options to treat serious infections caused by these microorganisms [1], which are also associated with higher economic costs. For this reason, the World Health Organization (WHO) listed CRE and ESBL-producing Enterobacterales as top critical-priority pathogens for which new therapeutic options and surveillance are urgently needed [2].

Carbapenem-resistance is normally mediated by the production of carbapenemases, which are commonly associated with highly prevalent international clonal groups [3]. Among carbapenemases, New Delhi metallo-β-lactamases (NDM) have been identified worldwide, and their prevalence has increased substantially in recent years in South America [4,5]. Worryingly, infections produced by NDM-producers are associated with high mortality rates, representing an important challenge for public health [5]. Among NDM carbapenemases, NDM-1 has been identified in several Gram-negative pathogens, including Enterobacterales and non-fermenting rods recovered in different geographical areas, from human and animal hosts and environmental sources [5,6].

Due to the emergence of carbapenem-resistant bacterial pathogens mediated by NDM-enzymes, Wu et al. published a comprehensive review on NDM enzymes that included the epidemiological data of NDM-positive Enterobacterales deposited in GenBank in 2018. Accordingly, the authors demonstrated that most countries from Latin America (LA) showed a prevalence of ≤5%, whereas the authors did not find data in some countries, such as Chile and Bolivia [5]. In this sense, the Reference Laboratory of Healthcare-Associated Infections (HAIs) of the Instituto de Salud Pública de Chile (ISP, Chile) confirmed the first isolation of the *K. pneumoniae* producer of NDM-1 in Chile in May 2014 (https://www.ispch.cl/sites/default/files/BoletinRam-30112015A_0.pdf, accessed on 5 March 2022). Importantly, we previously determined that this isolate, named UCO-361, was genetically related to an NDM-producer *K. pneumoniae* collected in Brazil, revealing the importance of genomic surveillance [7]. After their first identification, 84 NDM-producing Gram-negative isolates were reported during the 2014–2017 period in Chile (https://www.ispch.cl/sites/default/files/BoletinCarbapenemasas-02042019A%20(1).pdf, accessed on 5 March 2022), which demonstrates that these enzymes are circulating in the country at an unknown frequency. Because of this phenomenon, it is highly important to determine the role of plasmids, such as megaplasmids, in the rapid dissemination of carbapenemases. In this sense, *bla*_NDM_ gene is contained in plasmids of different replicon types, where IncX3 and IncC (IncA/C2) have been identified as the most prevalent type of plasmids carrying these genes in Enterobacterales worldwide [5,8]. Recently, diverse megaplasmids have been associated with the dissemination of multidrug resistance in clinically relevant pathogens [9,10,11,12]. Megaplasmids are referred to large plasmids of more than 100-kp that are characterized by their mosaic structure containing regions from diverse origins, including metals- and/or antibiotic-resistance genes (ARGs) [12,13]. In this study, we used a genomic approach to characterize a carbapenem-resistant *Klebsiella pneumoniae* (UCO-361 strain) clinical isolate carrying a novel transferable NDM-1-encoding megaplasmid.

## 2. Materials and Methods

A carbapenem-resistant *K. pneumoniae* strain, UCO-361, that corresponds to the first NDM-1-carrying strain detected in Chile, was analyzed. UCO-361 was recovered from a rectal swab of a colonized inpatient in a public hospital in Santiago (Capital of Chile) and is genetically related to a Brazilian clone [7]. This isolate was recovered from a hospitalized patient that shared a room with a colonized patient who was previously hospitalized in Brazil [7]. Moreover, *Escherichia coli* J53 (sodium-azide-resistant strain, SAZ^R^) was used as a recipient strain for mating experiments. *E. coli* ATCC 25922 and *Serratia marcescens* UCO-143 (*bla*_IMP_^+^) strains were used as controls for susceptibility studies and as a positive control for carbapenemase production in phenotypic tests, respectively. The strains *Acinetobacter baumannii* UCO-323 and *A. baumannii* UCO-324, which were previously characterized in our laboratory by the detection of *bla*_NDM-1_ by conventional PCR and Sanger sequencing, were used as positive controls for the *bla*_NDM-1_ gene.

Susceptibility studies were carried out by disk diffusion and broth dilution methods, according to the recommendations of Clinical & Laboratory Standards Institute (CLSI) [14]. The following antimicrobial disks (Oxoid Ltd., Basingstoke, UK) were included: imipenem (IPM, 10 μg), meropenem (MEM, 10 μg), ertapenem (ETP, 10 μg), cefotaxime (CTX, 30 μg), ceftazidime (CAZ, 30 μg), cefepime (FEP, 30 μg), amoxicillin/clavulanate (AMC, 20/10 μg), piperacillin/tazobactam (TZP, 100/10 μg), ceftazidime/avibactam (30/20 μg), aztreonam (ATM, 30 μg), amikacin (AMK, 30 μg), gentamicin (GEN, 10 μg), levofloxacin (LEV, 5 μg), ciprofloxacin (CIP, 5 μg), tetracycline (TET, 30 μg) and sulfamethoxazole/trimethoprim (SXT, 23.75/1.25 μg). Moreover, susceptibility to colistin (CST) was determined by the colistin-agar test (CLSI, 2021). Additionally, Carba NP method was used to test the carbapenemase activity [14], whereas metallo-betalactamase (MβL) production was investigated by the imipenem-EDTA disk assay [15].

Initial detection of *bla*_NDM-1_ was carried out by conventional PCR, using primers and conditions described by Poirel et al., 2011 [16]. Next, UCO-361 was subjected to whole genome sequencing (WGS) through short- (NextSeq, Illumina Paired-end platform) and long-reads (MinION; Oxford Nanopore Technologies). For the transconjugant strain (*E. coli* Tc-01), short-read (Illumina) sequencing was utilized to confirm the antibiotic-resistance genes (ARGs) transferred after mating assays. De novo assemblies were achieved using Unicycler v0.4.8 and SPades v3.15.4 assemblers, depending on the sequencing platforms utilized [17]. The resulting contigs were visualized with Bandage v0.8.1, whereas genomic annotation was performed using Prokka [18].

From the assembled genome, multilocus sequence type (MLST), plasmid replicons, antimicrobial resistance genes, virulence genes, capsule (K) and lipopolysaccharide (O) loci were identified using the platforms MLSTFinder 2.0 (https://cge.food.dtu.dk/services/MLST/, accessed in 5 March 2022), PlasmidFinder 2.1 (https://cge.food.dtu.dk/services/PlasmidFinder/, accessed in 5 March 2022), ResFinder 4.1 (https://cge.food.dtu.dk/services/ResFinder/, accessed in 5 March 2022), and PathogenWatch (https://pathogen.watch/, accessed in 5 March 2022), respectively. Assembly, annotation, and visualization of plasmids were carried out by Proksee (https://proksee.ca/, accessed in 5 March 2022). Plasmid sequences were compared against the PLSDB plasmid database [19].

Conjugation assays were conducted as described by Potron et al. [20]. For transconjugant selection, we utilized tryptic soy agar (Oxoid Ltd., Basingstoke, UK) plates containing SAZ (300 μg/mL) and supplemented with different antibiotics combinations per plate: SAZ-AMP (16 μg/mL), SAZ-ETP (0.125 μg/mL), SAZ-AMK (4 μg/mL) and SAZ-CTX (4 μg/mL). Mating experiments were carried out at 27 and 37 °C, and conjugation frequency (R) was estimated as R = number of transconjugants/number of recipients [21]. To verify the isogenicity between the recipient and transconjugant strains, they were subjected to Enterobacterial Repetitive Intergenic Consensus PCR (ERIC-PCR) according to Aydin et al. [22].

## 3. Results and Discussion

UCO-361 was isolated from a rectal swab sample from a colonized inpatient. It displayed resistance to IPM, MEM, ETP, CTX, CAZ, FEP, AMC, TZP, CZA, ATM, GEN, TET, SXT and CIP, intermediate to LEV and susceptible to AMK and CST (MIC < 3.5 µg/mL). According to the definition proposed by Magiorakos et al. [23], UCO-361 corresponds to an extensively drug-resistant (XDR) strain. Moreover, Carba NP and imipenem-EDTA tests were positive for the donor (UCO-361) and transconjugant (Tc-01) strains, evidencing the presence of a transferable MβL. Moreover, PCR results revealed the presence of *bla*_NDM_ in UCO-361.

To characterize the genetic features of UCO-361, we performed a hybrid analysis using short-read (Illumina, San Diego, CA, USA) and long-read (Oxford Nanopore Technologies, Oxford, UK) sequencing platforms. As a result, MLSTFinder analysis revealed that UCO-361 belonged to the ST1588, which has been previously identified in 3 hospitals in Rio de Janeiro, Brazil [24]. Moreover, UCO-361 presented the capsular serotype KL108/O1 that has been associated with hypervirulent clones [25]. Specifically, multidrug-resistant and hypervirulent strains associated with KL108 have been detected in different regions [25]. Furthermore, virulence genes associated with the hyperproduction of capsule (hypermucoid phenotype), such as *rmpADC* and *rmpA2* [25], were not detected in UCO-361.

On the other hand, ResFinder was used to detect the ARGs contained in UCO-361, which encompassed several β-lactamases, including ESBLs (*bla*_CTX-M-15_ and *bla*_SHV-106_) and carbapenemase (*bla*_NDM-1_), aminoglycosides-modifying enzymes (*aph(3″)-lb, aac(3)-lla, aph(6)-ld, aac(6′)lb-cr*), and quinolones- and trimethoprim/sulfamethoxazole-resistance genes (Table 1). Moreover, *bla*_NDM-1_ was detected in Tc-01 in addition to other β-lactamases, aminoglycosides-modifying enzymes, and quinolones-resistance genes (Table 1).

From the hybrid-sequencing approach, we identified a megaplasmid of 314,976 bp in UCO-361, named pNDM-1_UCO361, that carries the *bla*_NDM-1_ and *oqxB* genes, as the ARG detected (Figure 1A). Furthermore, we determined that *bla*_NDM-1_ was in the Tn*3000* transposon that includes a copy of IS*3000* and truncated ΔIS*Aba125* upstream *bla*_NDM-1,_ whereas the bleomycin-resistance gene *ble*_MBL_, in addition to *trpF*, *dsdD*, Δ*groES*, *groEL* and a copy of IS*3000,* were located downstream (Figure 1B). In this sense, *bla*_NDM_ genes are frequently described surrounded by the insertion sequence (IS) IS*Aba125* upstream *ble*_MBL_ downstream [5,8], which is congruent with our results. Interestingly, *bla*_NDM-1_ has been previously detected in a plasmid within the transposon Tn*3000* [24], remarking the relevance of this transposon in the spread of NDM-1.

Remarkably, pNDM-1_UCO361 does not match with any Inc group deposited in the PlasmidFinder database. Moreover, comparisons of pNDM-1_UCO361 using the PLSDB platform (https://ccb-microbe.cs.uni-saarland.de/plsdb/, accessed on 5 March 2022) showed no matches with complete plasmid sequences retrieved from the NCBI nucleotide database. Furthermore, using the mash dist function of PLSDB, we determined that the closest plasmid related to pNDM-1_UCO361 corresponded to the pNDM-1-EC12 plasmid isolated in *Enterobacter cloacae* strain EC12 (accession number NZ_MN598004.1), in which *bla*_NDM-1_ was identified in a common region of 2488 bp. Moreover, *bla*_NDM-1_ has been previously identified in a megaplasmid (ca. 350 kb) in *Raoultella ornithinicola* collected from water samples in rural China [11]. This plasmid (accession number CP041388) contained the *bla*_NDM-1_ in a different genetic environment in comparison to pNDM-1_UCO361 since it was bracketed by two copies of IS*91*, conforming to an unusual integron class 1 [11].

In terms of conjugation-related proteins, pNDM-1_UCO361 contained the *tra*C gene (Figure 1A), which encodes for a type-IV secretion system involved in the assembly of the F conjugative pilus [26]. Furthermore, pNDM-1_UCO361 presented several mobile-genetic elements, such as insertion sequences (IS) (Figure 1A), which remarks its potential to acquire exogenous genes. In addition to *tra*C, pNDM-1_UCO361 carries the *hns* gene that encodes for the H-NS protein, which regulates the expression of T4SS during plasmid conjugation [27].

Interestingly, although pNDM-1_UCO361 harbors the *tra*C gene, we detected an additional plasmid of 197,209 bp, which belongs to the IncFIB(K) containing the complete *tra* locus, which is involved in plasmid conjugation [28]. Moreover, this IncFIB(K) plasmid does not carry any antibiotic-resistance gene; thus, we hypothesized that it might mediate the transfer of pNDM-1_UCO361 simultaneously with other plasmids carrying additional ARGs. In this sense, we were able to transfer the *bla*_NDM-1_ gene through the mating assay between *K. pneumoniae* UCO-361 and *E. coli* J53 only at 27 °C in both agar and broth media. Moreover, we determined that the conjugation frequency of the *bla*_NDM-1_ corresponded to 4.3 × 10^−6^ transconjugants/recipient cell. As a result, the transconjugant *E. coli* strain Tc-01 displayed the same ERIC-PCR banding pattern with strain J53 (recipient), demonstrating their isogenicity.

Phenotypically, Tc-01 exhibited resistance to IPM, MEM, ETP, CTX, CAZ, FEP, AMC, TZP, ATM, AMC, CZA, GEN, TET and non-susceptibility to CIP, LEV, AMK and CST. From WGS data, we determined that Tc-01 obtained several ARGs (Table 1), which explains the resulting multidrug-resistant resistant phenotype displayed by this strain. As expected, the *bla*_SHV-1_ gene was not transferred from UCO-361 due to its chromosomal location in *K. pneumoniae* [29]. These results suggest that additional genes and/or plasmids were co-transferred with pNDM-1_UCO361, conferring resistance to various antibiotics. Unlike previously published studies [30,31], both donor and transconjugant strains were resistant to CIP, which is congruent to the increasing prevalence of plasmid-borne quinolone-resistance genes reported previously in Chile [32] and is consistent with what was reported in Brazil [24]. Furthermore, we determined that the IncFIB(K) plasmid was not transferred after the conjugation assay since the replicon was not detected in the whole genome of Tc-01.

Importantly, conjugation was effective only at 27 °C, with frequencies higher than that previously reported at 25 °C [31,33]. The successful transfer of the *bla*_NDM-1_ gene at this temperature might be related to its environmental origin [5,34]. In this regard, temperature could play a fundamental role in the transfer of *bla*_NDM-1_, and even other resistance genes, due to the thermosensibility of the conjugation apparatus in some plasmid families. Accordingly, it has been established that the *tra*G gene, which is present in the IncFIB(K) plasmid in UCO-361, encodes a binding protein between the relaxosome and the mating pair complex in the conjugation process. This gene is transcribed in a temperature-dependent manner, where its expression levels are considerably reduced at 37 °C [35]. Moreover, Gibert et al. demonstrated that the conjugation process R27 plasmid is thermoregulated, being promoted at 25 °C and suppressed at 37 °C [36]. However, the *hns* gene, which is present in pNDM-1_UCO361, has been described as inhibiting the conjugation of IncX3 plasmids at 30 and 42 °C and not at 37 °C [27]. Hence, we hypothesize that the IncFIB(K) plasmid present in UCO-361 plays a relevant role in the transfer of pNDM-1_UCO361, independently of the inhibition activity of H-NS. Therefore, it is important to elucidate the role of temperature in the regulation of proteins involved in the transfer of resistant megaplasmids since the likelihood of resistance gene transfer may be more frequent in the environment than in the human body.

In conclusion, it is important to evaluate the role of resistance plasmids that can contribute to the emergence of carbapenem-resistance Gram-negative bacteria. Although it is well known that the spread of NDM-1 occurs in the environment, it is necessary to understand the effect of temperature and plasmid interactions on this phenomenon.

## Figures and Tables

**Figure 1 antibiotics-11-01207-f001:**
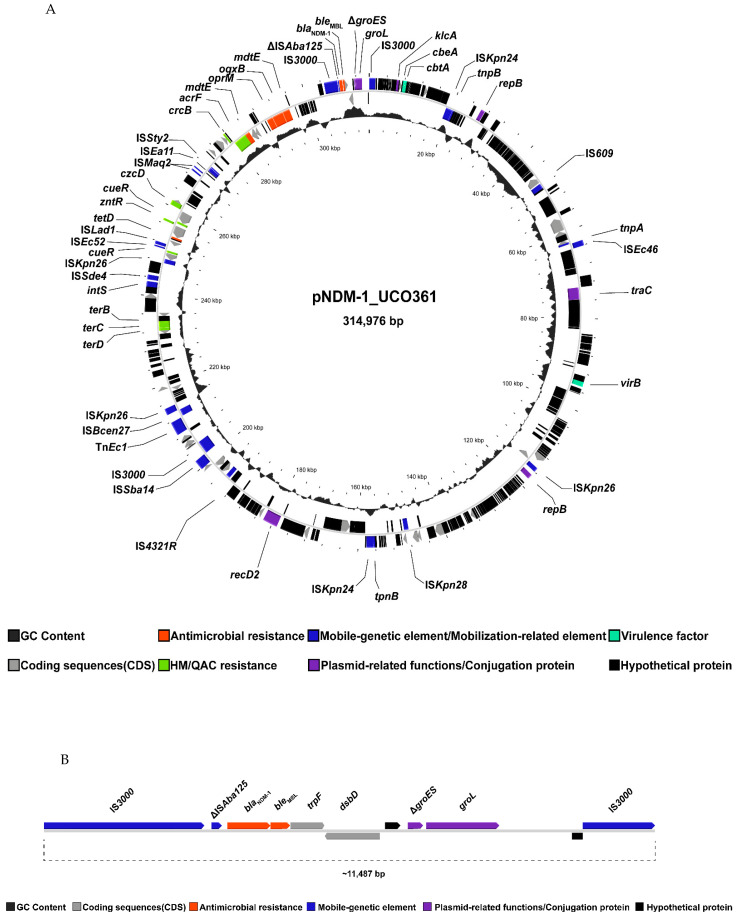
Circular representation of plasmid harboring *bla*_NDM-1_ in *K. pneumoniae* UCO-361. (**A**) General representation of genes encoded by pNDM-1_UCO361 showing coding sequences (CDS) in black, mobile-genetic elements in green, antibiotic-resistance genes in red and plasmid replication/conjugation proteins in blue. The genetic context of *bla*_NDM-1_ is highlighted inside the red box. (**B**) Linear view of Tn*3000* containing the *bla*_NDM-1_ gene. Figures were created with Proksee (Proksee.ca).

**Table 1 antibiotics-11-01207-t001:** Susceptibility profile and resistome of *K. pneumoniae* UCO-361 and transconjugant *E. coli* Tc-01 strains.

Strains	MIC (µg/mL)							Resistome			
	IPM	MEM	ETP	ATM	CTX	AMK	CIP	β-lactams	aminoglycosides	quinolones	SXT
*K. pneumoniae* UCO-361	16	>32	32	>32	>512	8	2	*bla*_NDM-1;_*bla*_SHV-106_; *bla*_CTX-M-15_; *bla*_OXA-1_; *bla*_TEM-1B_	*aac(3)-IIa; aac(6′)-Ib-cr; aph(6)-Id; aph(3″)-Ib*	*qnrB1*; *oqxA*; *oqxB*; *aac(6′)-Ib-cr*	*dfrA14*; *sul2*

*E. coli* Tc-01	16	32	16	16	512	2	2	*bla*_NDM-1_; *bla*_CTX-M-15_; *bla*_OXA-1_; *bla*_TEM-1B_	*aac(6′)-Ib-cr aph(6)-Id*; *aph(3″)-Ib*	*qnrB1;* *aac(6′)-Ib-cr*	

*E. coli* J53	0.25	≤0.06	0.016	≤0.06	0.063	1	0.002	-	-	-	-

MIC, minimum inhibitory concentration. IPM, imipenem; MEM, meropenem; ETP, ertapenem; ATM, aztreonam; CTX, cefotaxime; AMP, ampicillin; AMK, amikacin; CIP, ciprofloxacin; SXT, trimethoprim/sulphamethoxazole.

## Data Availability

This Whole Genome Shotgun project has been deposited at DDBJ/ENA/GenBank under the accession JAMJQY000000000 (WGS) and JAMJQY010000002.1 (plasmid pNDM-1_UCO361). The version described in this paper is version JAMJQY010000000.

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
