# Peer review of "Novel Megaplasmid Driving NDM-1-Mediated Carbapenem Resistance in Klebsiella pneumoniae ST1588 in South America"

_antibiotics, 2022, doi:10.3390/antibiotics11091207_

Round 1

Reviewer 1 Report

REVISION FOR  antibiotics-1871429

1)     In the introduction you should end with the paragraph Lines 210-225 appeared in the Discussion

2)     You would better have both plasmids of interest IncFIB(K) and pNDM-1_UCO361 in a separate transcojugant or transformant clone to study them in detail.

3)     You should better test all possible conjugation temperatures before evaluating the conjugation frequency

4)     Conjugation inhibition assumption is not clear in the text.

Author Response

Dear Reviewer,

Thank you very much for your comments on our manuscript, we really appreciate it.

In order to answer your requirements:

  1. We re-organised the introduction section, in which we added the information included between lines 210-225.
  2. We strongly agree with your observation. However, when we attempted to transfer the blaNDM gene, all plasmids were co-transfered in one step. In this sense, we are considering this paper as a baseline for future investigations about the role of each plasmid and temperature in the transferring process. Our next aim in this field, is to study each plasmid in separate, in order to study them in detail, as you suggested.
  3. We carried out the conjugation experiments at 3 different temperatures: 27°C and 37°C (Line 123). In this sense, the transfer of blaNDM was succesful only at 27°C.
  4. We discussed the fact that pNDM-1_UCO361 was transferred at specifically at 27°C but not at 37°C. As potential reasons for this phenomenon, we added previous evidence about the role of certain genes in the regulation of conjugation in a temperature-dependant manner. This information is included in lines 221-225. However, due to the lack of robust data, we just hypothetised that conjugation was inhibited at 37°C to the role of traG. The general action of traG is explained between lines 219-221.

As general observations, this is a general description of a plasmid that can be mobilized at 27°C, similar to common environmental temperatures, and inhibited at 37°C. It corresponds to the baseline that will allow us to study further the role of temperature in the dissemination os plasmids containing carbapenemases genes.

Reviewer 2 Report

The article describes a novel conjugative megaplasmid driving NDM-1 in Klebsiella pneumoniae, which may contribute to the spread of this gene. Other megaplasmids carrying antimicrobial resistant genes (AMR) have been previously described, including a non-conjugative one with NDM-1.  

Minor changes:

Line 58: Put references 12 and 13 in order.

Line 65: Although epidemiological information about the patient is documented in reference 14, it may be convenient to include it in the article.

Lines 67-71: Please, describe how the UCO-143 and UCO-323 strains were characterized.

Line 122:  Were other hypervirulent genes (rmpA, magA) detected in UCO-361? It should be specified in the text, as “associated to hypervirulent clones” does not provide enough information about this issue.

Line 129: Table 1: Please, note that it is a bit difficult to see which genes correspond to the different strains. Greater separation of the lines could provide a better understanding.

Line 133: “the blaNDM-1 gene as the unique ARG detected”: oqxB is a ARG. Besides being on the bacterial chromosome, it is also present on mobile insertion elements.

Line 155: Another megaplasmid containing NDM-1 has been described in Raoultella ornithynolitica (reference 11). It may be informative to compare both megaplasmids and analyze the encoded genes (as in figure 4 in reference 10)

Line 213: The abbreviation LA has not been previously defined.

Author Response

Dear Reviewer,

Thank you very much for your valuable comments and suggestions on our manuscript.

In relation to your comments:

  1. The references order was corrected (Line 75)
  2. Additional epidemiological information was added (Lines 82-85)
  3. We added the characterization of UCO323 and UCO324 (Lines 90-91)
  4. We added information about the relevance of KL108. Additionally, we screened for hypermucoid-related genes, which were not detected (Lines 143-146)
  5. The configuration of Table 1 was corrected.
  6. We included the oqxB gene in the description of ARGs present in the plasmid (Lines 157-158)
  7. We added a comment about the plasmid found in Raoultella ornithynolitica (Lines 173-177). Although in this work, blaNDM-1 was found in a megaplasmid, the structure of the genetic environment directly related to the gene is completely different. Due to this, we did not include a comparison figure since the general structures are very different.
  8. The LA abbreviation is explained in Line 55.